# Building Low-Cost, High-Performance Flexible Photodetector Based on Tetragonal Phase VO_2_ (A) Nanorod Networks

**DOI:** 10.3390/ma16206688

**Published:** 2023-10-14

**Authors:** Wenhui Lin, Chaoyang Tang, Feiyu Wang, Yiyu Zhu, Zhen Wang, Yifan Li, Qiuqi Wu, Shuguo Lei, Yi Zhang, Jiwei Hou

**Affiliations:** 1Department of Physics, School of Physical and Mathematical Science, Nanjing Tech University, Nanjing 211816, China; 13588028173@163.com (W.L.); tcyofficial@njtech.edu.cn (C.T.); fywang0907@163.com (F.W.); yiyuzhu@njtech.edu.cn (Y.Z.); zhenwang@njtech.edu.cn (Z.W.); 202161121005@njtech.edu.cn (Y.L.); 202021154023@njtech.edu.cn (Q.W.); sglei@njtech.edu.cn (S.L.); 2School of Energy Science and Engineering, Nanjing Tech University, Nanjing 211816, China

**Keywords:** tetragonal phase VO_2_ (A), low-cost, high-sensitivity photoresponse, flexible photodetector

## Abstract

We present a straightforward and cost-effective method for the fabrication of flexible photodetectors, utilizing tetragonal phase VO_2_ (A) nanorod (NR) networks. The devices exhibit exceptional photosensitivity, reproducibility, and stability in ambient conditions. With a 2.0 V bias voltage, the device demonstrates a photocurrent switching gain of 1982% and 282% under irradiation with light at wavelengths of 532 nm and 980 nm, respectively. The devices show a fast photoelectric response with rise times of 1.8 s and 1.9 s and decay times of 1.2 s and 1.7 s for light at wavelengths of 532 nm and 980 nm, respectively. In addition, the device demonstrates exceptional flexibility across large-angle bending and maintains excellent mechanical stability, even after undergoing numerous extreme bending cycles. We discuss the electron transport process within the nanorod networks, and propose a mechanism for the modulation of the barrier height induced by light. These characteristics reveal that the fabricated devices hold the potential to serve as a high-performance flexible photodetector.

## 1. Introduction

One-dimensional semiconducting nanostructures, such as nanowires, nanotubes, nanobelts, and nanoribbons, have captured considerable attention owing to their novel physical and chemical properties and potential applications in nanoscale electronics and optoelectronics [1,2,3]. They are characterized by a significant surface-to-volume ratio, a high aspect ratio, easy surface functionalization, and a unique electron-limited domain effect, which can extend the lifetime of photogenerated carriers [4]. Additionally, their linear geometrical structure provides them with good elasticity against external stresses, and they are less prone to cracking after deformation. These features make one-dimensional semiconductor nanowires ideal for designing and preparing high-performance photodetector devices. Many high-performance photodetectors utilizing one-dimensional semiconductor nanostructures, such as ZnO, Zn_2_GeO_4_, SnO_2_, and V_2_O_5_ have been prepared [5,6,7]. Among the variety of 1D semiconductor nanostructures, Vanadium dioxide (VO_2_) exhibits a reversible metal-insulator transition at 68 °C, where its crystal structure changes from a monoclinic phase at low temperatures to a rutile-type tetragonal phase at high temperatures, accompanied by significant changes in its electrical, magnetic, and optical properties. These novel properties enable a broad array of applications for VO_2_, including photodetectors, actuators, gas sensors, Mott-effect transistors and smart windows [8,9,10]. 

VO_2_ serves as a typical binary compound with multiple polymorphisms, such as VO_2_ (B), VO_2_ (A), VO_2_ (M1), and VO_2_ (R) [11,12]. These distinct polycrystalline structures have different properties and many different applications. For instance, due to its layered crystal structure, VO_2_ (B) is often used as a potential high-performance lithium battery electrode material in energy storage applications. Due to the robustness and durability of the structure and the unique open channels for Li-ion conduction, epitaxially grown VO_2_ (B) films can reach the theoretical limit of capacity through its sustained charge/discharge cycling capability [13]. VO_2_ (M1) and VO_2_ (R) have reversible metal-insulator phase transition properties and high temperature resistance coefficients, making them suitable for infrared photodetectors, optical switches, and smart windows [14,15]. Over the past few decades, substantial research efforts have been directed towards the comprehensive investigation of the syntheses, features, and potential applications of VO_2_ (B) and VO_2_ (M1), VO_2_ (R). However, there have been fewer studies of other polymorphs in the VO_2_ system that may also include unique physical and chemical characteristics as well as potential applications. Tetragonal phase vanadium dioxide, known as VO_2_ (A), exhibits a comparable reversible phase transition around 162 °C; however, its electrical and optical properties have been sparsely reported due to insufficient attention [16]. The crystal structure of VO_2_ (A) is constructed from interconnected three-dimensional arrays of VO_6_ octahedra. It is noteworthy that the V-O spacing in the [110] orientation is smaller compared to the other directions, which favors its preferential growth along the one-dimensional direction. Several studies have recently been carried out on the physical properties and structure of tetragonal phase VO_2_ (A). Belt-shaped VO_2_ (A) with a rectangular profile was produced and exhibited good thermal stability in air [17]. Both theory and experiments have revealed that VO_2_ (A) nanowires can be employed as H_2_S gas sensors, showcasing exceptional selectivity and reproducibility [18]. The structural evolution of VO_2_ (A) NRs under pressure reveals information about the VO_2_ (A) phase transition [19]. Through an analysis of the electrical properties of VO_2_ (A), its distinctive field emission characteristics have been elucidated [20]. VO_2_ (A) serves as a semiconductor material with a suitable optical band gap. However, a comprehensive study of the complex optical and electrical properties of nanostructured VO_2_ (A) remains a task to be undertaken. Consequently, the advancement of optoelectronic devices based on these materials requires an improved understanding of their photoresponse properties, and we herein investigate the optoelectronic characteristics of VO_2_ (A) NR networks utilized in a photodetector. 

The performance of a photodetector depends on the properties of its active material, including factors such as the absorption coefficient, bandgap width and morphology. Typically, multifunctional device fabrication involving nanostructures requires specialized substrate growth, which often requires a complex design process [21]. In this study, we demonstrate a low-cost flexible photodetector constructed with tetragonal phase VO_2_ (A) NR networks, which exhibits highly sensitive photoresponse properties (rise and decay times of 1.8 and 1.2 s, respectively). The device can be repeatedly bent at large angles without experiencing a significant loss of photoresponse properties. We have explored the electron transfer process in NR networks, revealing the mechanism of photocurrent generation and the factors affecting the performance. These features indicate that the fabricated flexible devices have the potential to be used as high-performance photodetectors. 

## 2. Experimental

### 2.1. Synthesis of Tetragonal Phase VO_2_ (A) Nanorods

Tetragonal phase VO_2_ (A) NRs were prepared through a hydrothermal method utilizing VOSO_4_ as the precursor material, and the synthesis method was based on our previous report [11]. VOSO_4_ was analytically pure and purchased from Shanghai Aladdin Biochemical Technology Co., Shanghai, China. Normally, VOSO_4_ powder (0.08 g) was dispersed in 40 mL of deionized water (deionized water with a resistivity of 18.2 MΩ · cm was prepared in the laboratory), forming a clear blue solution after vigorous stirring. Subsequently, the blue solution was transferred into a 50 mL Teflon-lined autoclave housed within a stainless-steel shell and maintained at 220 °C for 24 h. After cooling the solution to ambient temperature, a brown slurry precipitate formed. After centrifugation, the resulting precipitate was washed repeatedly with deionized water and ethanol. Ethanol was analytically pure and purchased from Shanghai Aladdin Biochemical Technology Co. This resulting precipitate underwent an overnight vacuum drying process at 60 °C for 8 h. Stable tetragonal phase VO_2_ (A) NRs were obtained in the final experiment. No organic surfactants were introduced during the whole experiment. 

### 2.2. Materials Characterization

The crystal structures of the acquired samples were analyzed through X-Ray diffraction (XRD) analyses conducted using a Smart-Lab diffractometer (Rigaku, Akishima, Japan) with a Cu-Kα radiation source. The microstructures of samples were examined using a field emission scanning electron microscope (SEM, Hitachi S-4800, Tokyo, Japan). Transmission electron microscopy (TEM) images, including high-resolution TEM (HRTEM) images and selected area electron diffraction (SAED) patterns, were obtained for the samples using a TEM system (JEOL-2010, Akishima, Japan) operated at an acceleration voltage of 200 kV. Absorbance spectrum was obtained using a UV-VIS-NIR spectrophotometer (Shimadzu, UV-3700, Tokyo, Japan).

### 2.3. Fabrication of Tetragonal Phase VO_2_ (A) Nanorod Networks Photodetector

During device fabrication, conventional UV lithography was used, followed by the deposition of gold electrodes (60 nm thick) on a polyethylene terephthalate (PET) substrate. Finally, the lift-off method was utilized to define the contact electrodes. The distance between the two gold electrodes measured approximately 15 μm, and each gold electrode had a length of about 2 cm. Tetragonal phase VO_2_ (A) NRs were suspended in deionized water using the ultrasonication process. The usual concentration maintained for this suspension was 0.2 mg/mL. The above solution was carefully deposited onto the pre-cleaned electrodes and allowed to air dry at ambient temperature. The device has been formed by a single-crystal VO_2_ (A) NR networks. The electrical characteristics of the devices were carefully characterized under ambient conditions using a two-probe structure. To assure the precision and reliability of the measurements, a Keithley 4200 (Tektronix, Beaverton, OR, USA) semiconductor analyzer was used. 

## 3. Results and Discussion

The scanning electron microscopy (SEM) analysis was conducted with a Hitachi S-4800 instrument to explore the morphology and microstructure of VO_2_ (A) NRs. A typical scanning electron microscope image of tetragonal phase VO_2_ (A) NRs is shown in Figure 1a, exhibiting lengths ranging in several tens of micrometers and widths of approximately 200 nm. The inset in Figure 1a is an SEM image at high magnification, providing a clearer view of the microstructure of the sample. The tetragonal phase VO_2_ (A) exhibits a distinct rod-like morphology. High-resolution transmission electron microscopy (HRTEM) imaging, utilizing a JEOL-2010 instrument, confirms the single-crystalline quality of the NRs, as shown in Figure 1b. Different lattice fringes can be observed, with spacing between the neighboring fringes measuring 0.6 nm and 0.19 nm, effectively matching the spacing between the (110) and (004) lattice planes. This result is consistent with previous reports in the literature [22]. The inset of Figure 1b demonstrates a selected area electron diffraction (SAED) pattern characterized by a clear diffraction dot matrix structure, further confirming that the tetragonal phase VO_2_ (A) NR is a single-crystal structure. Figure 1c shows the schematic diagram of the experimental configuration for performing the photoresponse measurements. 

VO_2_ (A) NRs were dispersed on the PET substrate and formed a network structure connecting two gold electrodes. The dark currents and photocurrents resulting from vertically incident illumination were measured using a two-probe technique. The construction of our devices presents the advantages of simplicity, high performance, and cost-effectiveness in contrast to expensive and complex electron beam lithography. Figure 1d shows a photograph of a typical device, which consists of a dispersed VO_2_ (A) NR network and two gold electrodes. The VO_2_ (A) NRs are uniformly distributed in the gap between the gold electrodes, which have a width of 15 μm and a length of 2 cm, respectively. 

The optical absorption spectrum of the tetragonal phase VO_2_ (A) NR was recorded at room temperature utilizing a spectrophotometer (Shimadzu UV-3700). The UV-Vis-NIR absorption spectrum of the tetragonal phase VO_2_ (A) NR is shown in Figure 2a. The optical absorption spectrum of the tetragonal phase VO_2_ (A) NR shows a distinct absorption peak at 2.3 eV (538 nm). The current–voltage (I–V) characteristic curve is obtained by applying a bias voltage from −4.0 V to 4.0 V, as shown in Figure 2b. The photocurrent measurements were performed under laser irradiation at wavelengths of 532 nm and 980 nm, with laser power densities of 85 and 120 mW/cm^2^, respectively. The photocurrent increased significantly under light, especially at high voltage bias. The I–V curves of the devices showed linear behavior both in the absence and presence of light, indicating that the contact between the tetragonal phase VO_2_ (A) NR and the gold electrode exhibits ohmic characteristics [23]. This indicates effective connectivity between the gold electrodes and the VO_2_ (A) NR networks, with the junction resistance being comparatively smaller than the total device resistance. The dynamic photocurrent of the device under the irradiation of modulated visible and infrared light (532 nm and 980 nm) is shown in Figure 2c, where the modulation frequency of visible and infrared light is 10 seconds on and off. Under irradiation of visible and infrared optical power densities of 85 and 120 mW/cm^2^, respectively, and a bias voltage of 2.0 V, the photodetector exhibited characteristic switching effects. 

This results in two different current states: a low-current state (in dark) and a high-current state (under illumination). Under visible light (532 nm) irradiation, the photocurrent of the device rises rapidly from 2.1 μA to 42.6 μA, marking a remarkable 1928% increase. Similarly, under infrared light (980 nm) irradiation, the device photocurrent rapidly increases from 2.3 μA to 8.8 μA, resulting in a remarkable 282% increase. Subsequently, when the irradiation of the light source is turned off, the photocurrent drops sharply to its initial level. After the absorption of photons by the tetragonal phase VO_2_ (A) NRs, the electrons are transitioned from the valence band to the conduction band, increasing the concentration of the electrons and holes, which leads to a significant increase in the photocurrent. The currents in both low- and high-current states remained almost constant for 10 cycles, indicating a remarkable reversibility and stability. In comparison to the photodetectors utilizing VO_2_ NRs as reported previously, the fabricated devices exhibit exceptional stability and reproducible switching behavior [24,25]. The response speed is a critical property of a photodetector, indicating the speed of the transition between a low-current and high-current state. The response times were defined as the duration taken for the current to increase from 10% to 90% of the maximum photocurrent or decrease from 90% to 10% during the on/off cycle. As illustrated in Figure 2d, the response times (rise and decay times) under visible (532 nm) irradiation are 1.8 s for rise and 1.2 s for decay, while under infrared (980 nm) irradiation for the VO_2_ (A) NRs, they are 1.9 s for rise and 1.7 s for decay. The devices exhibit a consistently stable and reproducible light-switching behavior, distinguishing them from the previously reported optical switches based on VO_2_ (A) NRs. The rapid response can be attributed to the single-crystal structure of the individual VO_2_ (A) NRs and the short current path within the VO_2_ (A) NR networks. These properties allow the constructed photodetector to outperform previously reported VO_2_ photodetectors, which demonstrated rise and decay times lasting for several seconds [26].

For a more comprehensive analysis of the dynamic response, we investigated the dynamic behavior of the on/off cycle separately under visible and infrared irradiation, respectively. Figure 3a presents a detailed depiction of the response profile for the ‘on’ cycle, which was subsequently fitted using a stretched exponential function: (1)$$I=I_{0}(1-exp{\left(-t/\tau_{r}\right)}^{d})\;$$
here, $I_{0}$ is the steady state photocurrent, $\tau_{r}$ is the rise time constant, t is the time, and d is an exponent between 0 and 1, which encompasses the characteristics of the relaxation process [27,28]. According to the experimental data and fitting results, the visible (532 nm) and infrared (980 nm) light intensities are 85 mW/cm^2^ and 120 mW/cm^2^ at a bias voltage of 2.0 V. The fitting calculations show that $\tau_{r}$ and d are 0.96 and 0.72, and 0.08 and 0.12 s, respectively. In addition, we analyzed the decrease in the photocurrent from the maximum steady-state value to the minimum dark current when the irradiated light source is turned off (Figure 3b). A second-order exponential decay model was used to fit the data to this process: (2)$$I=Aexp(-t/\tau_{1})+Bexp(-t/\tau_{2})$$
herein, the constants A and B represent weighting factors, and $\tau_{1}$ and $\tau_{2}$ are relaxation time constants [29,30]. The existence of double time constants indicates the existence of two distinct relaxation pathways, with the contributions of these different relaxation pathways being distinguished by the corresponding weighting factors. Generally, the value of $\tau_{1}$ is less than $\tau_{2}$, thereby $\tau_{1}$ represents the faster path component (related to bulk processes), while $\tau_{2}$ represents the more gradual component (related to surface processes). Figure 3b displays the experimental data fitted according to Equation (2) under visible and infrared irradiation, where the values of A, B, $\tau_{1}$, and $\tau_{2}$ under visible irradiation (532 nm) are 48.3, 1.9, 0.52 s, and 3.53 s, respectively. Similarly, under infrared irradiation (980 nm), the values of A, B, $\tau_{1}$, and $\tau_{2}$ are 7.6, 0.6, 0.7 s, and 3.4 s, respectively. A comparison of the photoresponse properties of different semiconductor oxide materials is shown in Table 1. Figure 3c presents the current–voltage (I–V) curves of the photodetector under dark and light conditions (532 nm visible light at different power densities), demonstrating that the measured I–V curves are linear, thus indicating the presence of an ohmic contact between the VO_2_ (A) NRs and the gold electrode. Figure 3c illustrates the dark current–voltage (I–V) characteristics of the photodetector and under visible light (532 nm) irradiation at different power densities. The I–V curve displays a distinct linear behavior, indicating that the contact between the VO_2_ (A) NR network and the gold electrodes is an ohmic contact. This suggests that the Au electrodes established a solid connection with the VO_2_ (A) NR networks, and the junction resistance was comparably lower than the overall resistance of the device. The illumination led to a significant surge in the photocurrent, especially noticeable at higher voltage biases. This suggests that the gold electrode establishes an ohmic connection to the VO_2_ (A) NR networks and that the junction resistance is less than the overall resistance of the device. The VO_2_ (A) NR absorbs photons and excites electrons and holes, leading to a large increase in the carrier concentration, which induces a significant increase in the photocurrent, especially at higher voltage biases. At a bias voltage of 4.0 V, the photocurrent exhibited a remarkable increase of approximately 29-fold, rising from 3.8 μA in the dark to 114.4 μA under visible light irradiation (with a light irradiance of 106.2 mW/cm^2^). At the same bias voltage, the photocurrent is gradually enhanced with increasing light intensity, which is attributed to the increase in the free carrier concentration in the VO_2_ (A) NR. The photocurrent (I) exhibited a clear dependence on the incident light intensity (P) (as illustrated in Figure 3d). This relationship can be expressed as $I\propto P^{b}$, with the exponent b being influenced by the density of the trap states and is usually less than 1 [31]. In particular, when there is a high trap state density, the lifetime of the photogenerated carriers is shortened, and the photocurrent exhibits a linear departure corresponding to the incident light power. Fitting the experimental data resulted in I=0.89, a non-uniform value that suggests a multifaceted process encompassing electron/hole generation, recombination, and trapping, similar to observations in V_2_O_5_ nanorods and VO_2_ (M1) microwires [7,32]. 

To investigate the flexibility of the devices, we further fabricated flexible photodetectors using VO_2_ NRs on mechanically flexible substrates (PET) as shown in Figure 4a,b. The dynamic photocurrent of the device in the flat state under irradiation with different visible power densities is shown in Figure 4a. At a bias voltage of 2.0 V, the photocurrent increases significantly with the increasing power density of visible irradiation. The ratio of photocurrent to dark current increases from 8.5 times to 27.2 times as the visible irradiation power density increases from 28.3 mW/cm^2^ to 106.2 mW/cm^2^. Figure 4b illustrates the correlation between the dynamic photocurrent and visible irradiation power density in the bent state, with a bias voltage of 2.0 V. The photocurrent increases significantly with increasing visible light irradiation power density when the device is bent at a large angle. Compared to the flat state, the photocurrent magnitude of the device remains almost constant with only slight fluctuations in the bent state for the same visible power density irradiation. The fluctuations in the photocurrent of flexible photodetector devices, in comparison to rigid photodetector devices, can be attributed to the unavoidable rearrangement of the nanowires and contraction sliding between them during bending. Furthermore, the contact between the nanorods, the flexible substrate, and the metal electrodes leads to slight fluctuations in the photocurrent during bending. We also conducted tests on the optoelectronic performance of the flexible device under repeated bending. Even after undergoing numerous repeated bends, the properties of the devices scarcely changed significantly. This can be attributed to the solid contact (ohmic contact) between the closely spaced nanorods and the metal electrodes. These features indicate that our photodetector device has potential applications in the area of flexible electronics. 

To explore the surface electron transport mechanism more deeply, we have investigated the dynamic photocurrent of the tetragonal phase VO_2_ (A) NR networks under vacuum and atmospheric conditions. Figure 5a shows the I–V curves recorded in the dark. The significant enhancement of the photocurrent under vacuum conditions suggests that reducing the ambient pressure can increase the conductivity. Additionally, this confirms that the adsorption and desorption behavior of the NR surface plays an important role in the photoresponse process. Figure 5b shows the dynamic photocurrent of the device measured in vacuum and air at a bias voltage of 2.0 V and a visible irradiance power density of 85 mW/cm^2^. Under photoirradiation conditions, the photocurrent increases rapidly, from 4.5 to 34.3 μA (in vacuum) and from 2.1 to 20.4 μA (in air), and the photocurrent gains of the devices are 662% and 871%, respectively. Fitting the rise and decay of the dynamic photocurrent with Equation (1) (Figure 5c) and Equation (2) (Figure 5d) yields $\tau_{r}=0.06$ s (in vacuum) and $\tau_{r}=0.13$ s (in air), respectively. Therefore, the presence of oxygen in air accelerates the saturation of the photocurrent, indicating that the surface states of the electron trap are quickly filled with holes. The exponent d was 0.03 (in vacuum) and 0.09 (in air), with corresponding $\tau_{1}$ values of 1.735 and 0.16 seconds, and $\tau_{2}$ values of 37.202 and 0.597 s, respectively. As described above, $\tau_{1}$ represents rapid interband recombination within the bulk, while $\tau_{2}$ corresponds to surface processes, contingent upon the presence of surface oxygen. Commonly, the inherent potential generated by adsorbed oxygen on the surface of NRs promotes the separation of photogenerated electron–hole pairs, which leads to the accumulation of holes on the NR surface [38]. Therefore, the electron–hole recombination indirectly on the NR surface possesses a prolonged time constant. 

The photodetection mechanism of the VO_2_ (A) NR networks includes the production of free charge carriers and their subsequent transportation, encompassing interfaces between neighboring nanorods and the metal/VO_2_ (A) interface. The surface states of the device, including vacancies, surface defects, and dangling bonds in the VO_2_ (A) NRs, can affect the generation and quantity of free charge carriers. The adsorption effect of the surface states has been reported in many photodetector devices based on oxide semiconductor nanowire networks, such as ZnO nanodisk [39], Zn_2_GeO_4_ nanowire networks [6], and so on. The VO_2_ (A) NRs exhibit n-type semiconductor behavior, with oxygen vacancies contributing electrons to the conduction band. Thus, the fundamental photodetection mechanism of these nanorods involves the adsorption and desorption of oxygen molecules, which modulates the production of free charge carriers and electron transport. 

In the dark, absorbed oxygen molecules capture electrons on the surface of the nanorods, generating surface states with negative charges (or oxygen ions) [$O_{2}\left(g\right)+e^{-}\to O_{2}^{-}(ad)$], causing a depletion layer to form near the surface, which reduces the electrical conductivity. The electron–hole pairs $\left[hv\to e^{-}+h^{+}\right]$ generated due to visible light illumination are then separated by the surface electric field. The additional holes migrate toward the nanorod surface following the potential gradient and combine with oxygen, leading to oxygen desorption from the surface of the VO_2_ (A) NR $\left[h^{+}+O_{2}^{-}(ad)\to O_{2}(g)\right]$ (Figure 6a). This increases the quantity of the unpaired electrons, raises the carrier concentration, and reduces the width of the depletion layer, resulting in a notable increase in the photocurrent. According to the literature reports, the adsorption and desorption of oxygen molecules play a key in influencing the photoconductivity of metal oxide semiconductors, including Zn_2_GeO_4_, SnO_2_, and ZnO [40,41]. Furthermore, the nanorods network structure may have an additional conduction mechanism that is absent in single nanorod devices. The resistance between two intersecting nanorods mainly originates from the impedance of the NR–NR junction rather than the effect of the intrinsic resistance of the NR. This junction barrier located near the surface of the NRs results in a significant band bending (Figure 6b), and the charge carriers have to pass through the junction barrier. In NR network photodetectors, the NR–NR junction barrier must be overcome when the electrons are transported from one nanorod (NR) to another (NR) (as shown in Figure 6c). The barrier to electron transfer arises from the surface depletion layer, which can be reduced by light due to the increase in carrier density, thus effectively lowering the effective barrier height. Under illumination, the height of the barrier potential of the NR–NR junction decreases, making it easier for charge carriers to pass through the NR network, resulting in an increase in the photocurrent. The modulation of the barrier height induced by light has been observed in optoelectronic devices utilizing nanobelts and nanowire networks as well [42,43]. 

## 4. Conclusions

In conclusion, single crystalline tetragonal phase VO_2_ (A) NRs were synthesized using a hydrothermal method. We have demonstrated that tetragonal phase VO_2_ (A) NR networks can be utilized to construct efficient flexible photodetectors. This method is straightforward, cost-effective, and does not require a complex lithography process. The devices exhibit exceptional photosensitivity, reproducibility, and stability under ambient conditions. With a bias voltage of 2.0 volts, the device demonstrates a photocurrent switching gain of 1982% and 282% under irradiation with light at wavelengths of 532 nm and 980 nm, respectively. The devices show a fast photoelectric response with rise times of 1.8 s and 1.9 s and decay times of 1.2 s and 1.7 s for light at wavelengths of 532 nm and 980 nm, respectively. The fabricated devices exhibit rapid response and recovery times, attributed to the predominant carrier transmission through NR–NR junctions. The dynamic response behavior suggests a mechanism for the modulation of the barrier height induced by light. The results from multiple large-angle bending experiments demonstrate that the device possesses excellent flexibility characteristics. This simple photodetector fabrication technique provides a promising path for manufacturing low-cost, high-performance, large-scale flexible wearable devices based on nanomaterials. 

## Figures and Tables

**Figure 1 materials-16-06688-f001:**
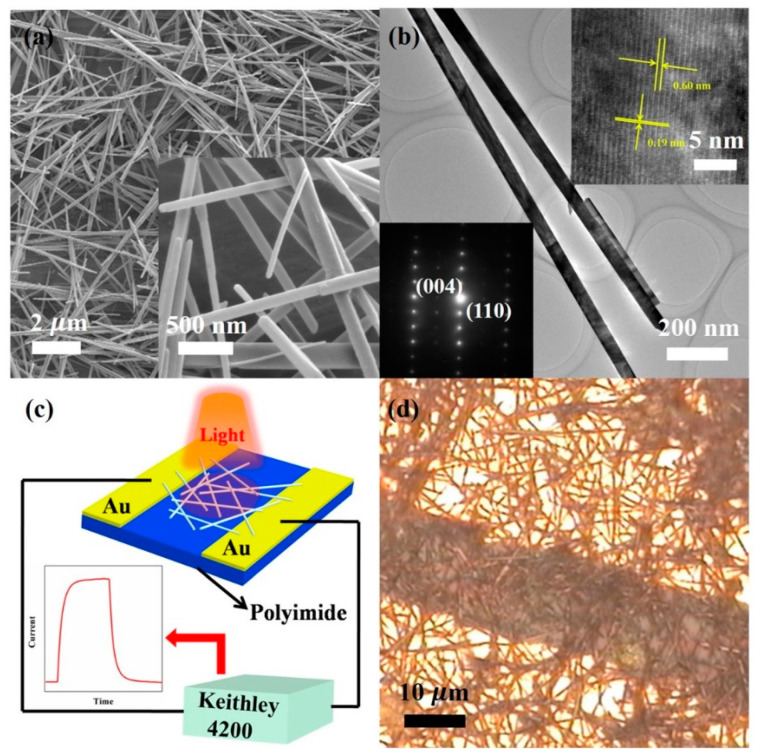
(**a**) Typical field-emission SEM (inset: high resolution SEM image, scale bar: 500 nm) image of tetragonal phase VO_2_ (A) nanorods, scale bar: 2 μm. (**b**) TEM image of tetragonal phase VO_2_ (A) nanorods (inset: the associated HRTEM image and SAED pattern, scale bar: 5 nm), scale bar: 200nm. (**c**) Schematic diagram of the tetragonal phase VO_2_ (A) nanorod network flexible photodetector. (**d**) Typical optical image of tetragonal phase VO_2_ (A) nanorods.

**Figure 2 materials-16-06688-f002:**
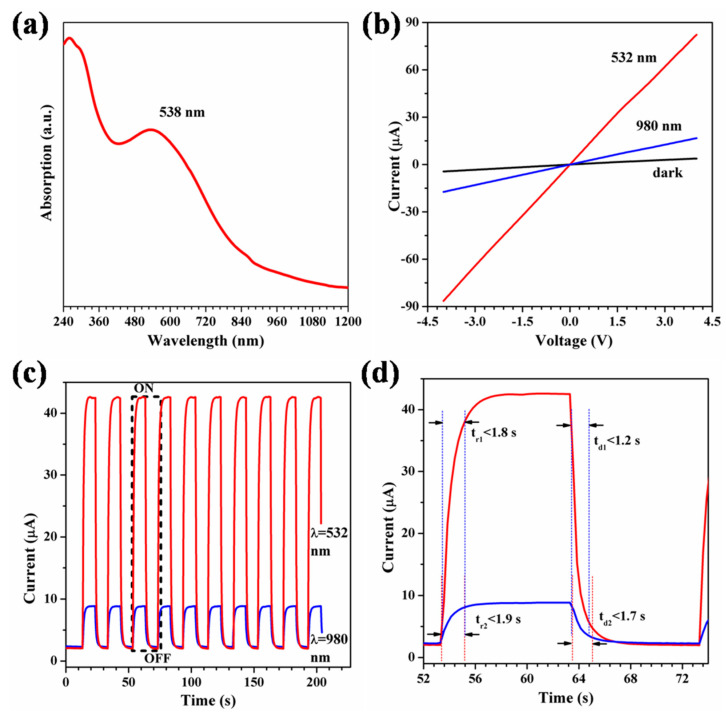
(**a**) UV-VIS-NR absorption spectrum of tetragonal phase VO_2_ (A) nanorods. (**b**) I–V curves measured in the dark and upon exposure to 532/980 nm illumination, the power densities of 532 nm and 980 nm light are 85 and 120 mW/cm^2^, respectively. (**c**) Time-dependent photoresponse under periodic illumination at 532 nm and 980 nm wavelengths with intervals of 10 s, applying a bias voltage of 2.0 V, the power densities of 532 nm and 980 nm light are 85 and 120 mW/cm^2^, respectively. (**d**) An enlarged view of the area enclosed by the black dashed line in (**c**).

**Figure 3 materials-16-06688-f003:**
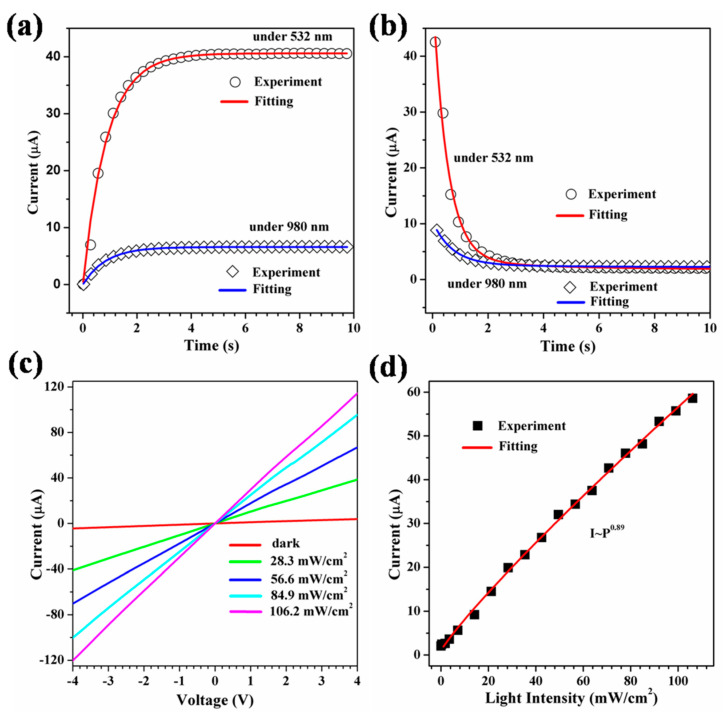
Fitting the (**a**) response curve and (**b**) reset curve under illumination at wavelengths of 532 nm and 980 nm, with a bias voltage of 2.0 V, the power densities of 532 nm and 980 nm light are 85 and 120 mW/cm^2^, respectively. (**c**) I-V curves of the device under 532nm illuminations with different light intensities. (**d**) Relationship between photocurrent and incident light density at a bias voltage of 2.0 V.

**Figure 4 materials-16-06688-f004:**
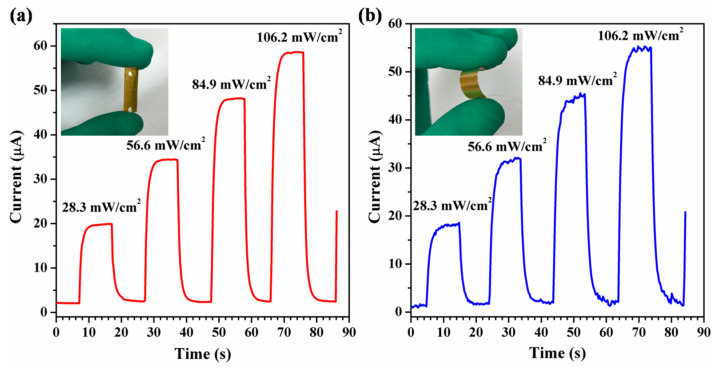
Time-dependent photoresponse to visible light (532 nm) at different irradiation intensities under periodic illumination at 10 s intervals, applying a bias voltage of 2.0 V, the inset image shows an optical picture of the test device. (**a**) Flat condition test. (**b**) Bending condition test.

**Figure 5 materials-16-06688-f005:**
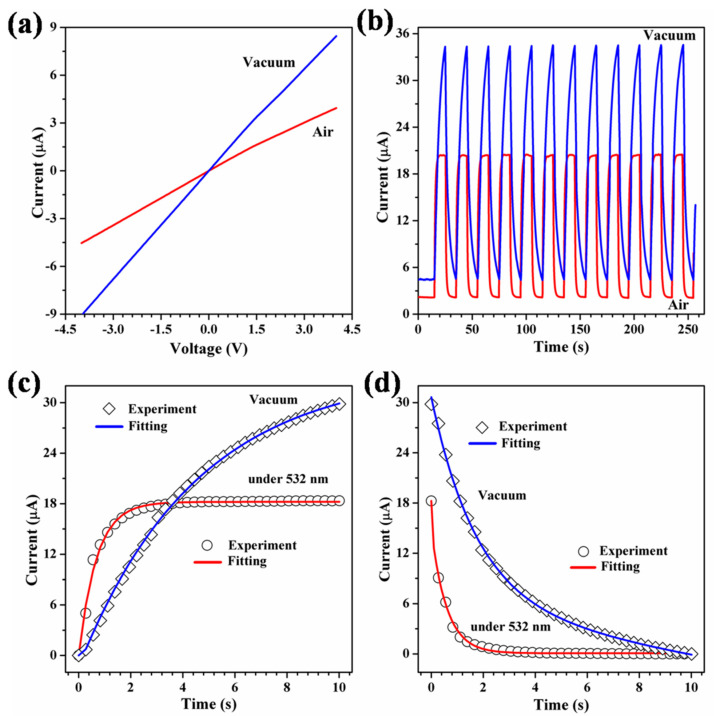
(**a**) I-V characteristics of the device in both air and vacuum in the dark. (**b**) Time-dependent photoresponse in both air and vacuum at 10 s intervals under periodic light illumination with a bias voltage of 2.0 V (light intensity 28.3 mW/cm^2^). (**c**) Fitting the response curve and (**d**) reset under light illumination in air and vacuum in the dark, applying a bias voltage of 2.0 V, the power densities of visible is 28.3 mW/cm^2^.

**Figure 6 materials-16-06688-f006:**
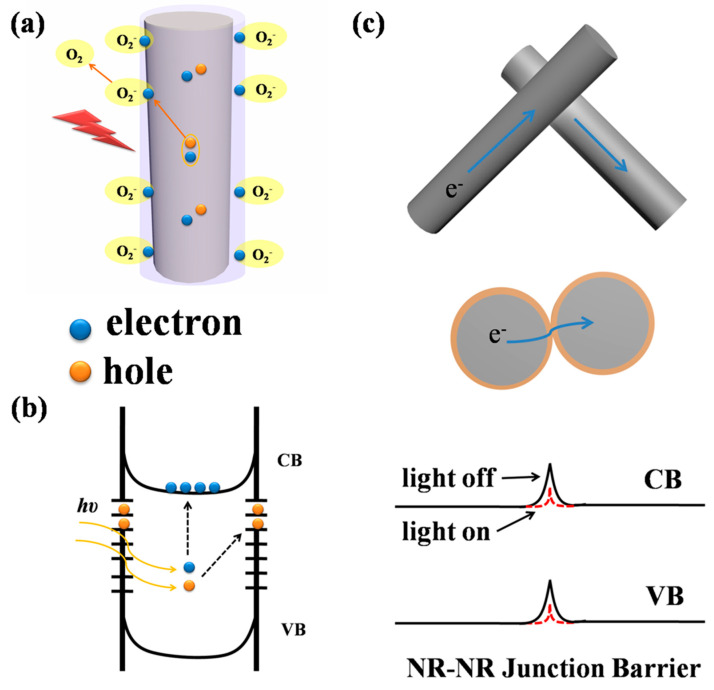
(**a**) Schematic depiction of oxygen adsorption in the dark and oxygen desorption upon nanorod illumination. (**b**) Excitation of hole and carrier jumps after photon absorption by nanorods. (**c**) Junction barrier for electron transport between nanorods (NR-NR) and its modification upon illumination.

**Table 1 materials-16-06688-t001:** The comparison of photoresponse properties for various semiconductor oxide materials.

Photodetector	Laser	$\tau_{r}$	$\tau_{d}$	Ref.
VO_2_ (M1) thin film	1550 nm	2.23 s	3.67 s	[33]
ZnO nanowires	365 nm	32 s	3.2 s	[30]
ZnO thin film	365 nm	24 s	15s	[34]
V_2_O_5_ nanorods	535 nm	0.79 s	0.54 s	[35]
SnO_2_ nanowires	250 nm	0.03 s	0.03 s	[36]
Sb_2_O_3_ networks	400 nm	0.3 s	0.3 s	[37]
Zn_2_GeO_4_ networks	254 nm	0.3 s	0.2 s	[6]
VO_2_ (A) NR networks	532 nm	0.96 s	0.52 s	this work
VO_2_ (A) NR networks	980 nm	0.08 s	0.70 s	this work

## Data Availability

Not applicable.

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
