# Peer review of "Building Low-Cost, High-Performance Flexible Photodetector Based on Tetragonal Phase VO2 (A) Nanorod Networks"

_materials, 2023, doi:10.3390/ma16206688_

Round 1

Reviewer 1 Report

This article reported the fabrication of a VO2-based photodetector. This article needs revision before being considered for publication.

1. Abstract and concussion must be more informative. The findings must be included. 

2. The introduction section must be updated. What types of materials are reported for similar applications? Advantages and disadvantages must be reported in materials, and VO2 must be discussed in detail. 

3. What is the novelty of the work?

4. Details of materials and chemicals used in this article must be added.

5. Authors focused on the Tetragonal Phase VO2 (A) Nanorod over other phases. What are the advantages of this phase over other phases? 

6. The language of the article must be improved. 

7. Results must be compacted with the literature. Authors can add a Table to compare the results and specific conditions. 

8. Results need More discussion. 

Need revision

Author Response

Dear Editor:

We thank reviewers for their time and constructive comments, which helped us much to improve the manuscript. We are extremely grateful to the reviewers for recognizing our work and carefully considered every comment, and made cautions revision accordingly. Based on these comments, we have answered the questions in detail one by one. All changes have been made in the revised manuscript as marked in blue color. As for these questions, the following blue words are our answers and discussions. If you have any other questions about this paper, I would quite appreciate it if you could let me know them. We look forward to hearing from you soon for a favorable decision. Thank you again for your time and consideration.

Most Sincerely

Jiwei Hou

Reviewer 2 Report

Report on paper “Building Low-Cost, High-Performance Flexible Photodetector 2 Based on Tetragonal Phase VO2 (A) Nanorod Networks” by Chaoyang Tang, Feiyu Wang, Yiyu Zhu, Zhen Wang, Yifan Li, Qiuqi Wu, Shuguo Lei, Yi Zhang, Jiwei Hou  

In the paper the authors consider synthesis of tetragonal phase VO2 (A) nanorod networks, employing a hydrothermal method described previously and documented. The authors give clear description of synthesis approach and properties of obtained materials by making use of various testing methods. Particular attention is paid for applications of such systems as photodetectors with favorable conclusions. The manuscript can be considered for publication in ‘J. Materials” as it contains relevant hints, new clear and transparent results.

Author Response

(The authors gave the same response as above.)

Reviewer 3 Report

This paper reports on a low-cost flexible photodetector fabricated using vanadium dioxide VO2 (A) NR networks, which exhibits highly sensitive photoresponse properties. The framework of this manuscript is clear and well-defined. I recommend it for publication after a major revision. Here are some comments.

1) Please include significant quantitative results also in the abstract.

2) In line 38, 'in characterized' -> is characterized?

3) SEM and TEM images in Fig. 1a and 1b should be scaled.

4) In line 220 and 228, is that 'Fig. 2b' is supposed to be referring to Fig. 3b. Please revise.

5) In line 269-270, 'This can be attributed to the solid contact between the closely packed nanowires and the metal electrodes.' How did the authors draw this conclusion?

6) Why bias voltage 2.0 V was specifically used for experimental results indicated in Fig.3d, Fig.4a and Fig.4b?

7) Suggest to cite relevant references to support line 306-313.

English is acceptable. Only minor correction is required.

Author Response

(The authors gave the same response as above.)

Round 2

Reviewer 1 Report

Acceptable

Reviewer 3 Report

Authors have sufficiently addressed all the comments and revised the manuscript accordingly. Therefore, I recommend it for publication.